# Individual experience as a key to success for the cuckoo catfish brood parasitism

Holger Zimmermann [1], Radim Blažek[1,2], Matej Polačik[1] & Martin Reichard [1,2,3 ✉]

Brood parasites are involved in coevolutionary arms races with their hosts, whereby adaptations of one partner elicit the rapid evolution of counter-adaptations in the other partner. Hosts can also mitigate fitness costs of brood parasitism by learning from individual or social experience. In brood parasites, however, the role of learning can be obscured by their stealthy behaviour. Cuckoo catfish (*Synodontis multipunctatus*) parasitise clutches of mouthbrooding cichlids in Lake Tanganyika and are the only non-avian obligate brood parasites among vertebrates. We experimentally demonstrate that cuckoo catfish greatly enhance their efficiency in parasitising their hosts as they learn to overcome host defences. With increasing experience, cuckoo catfish increased their parasitism success by greater efficiency through improved timing and coordination of intrusions of host spawnings. Hence, within the coevolutionary arms races, brood parasites learn to overcome host defences during their lifetime.

[1] Institute of Vertebrate Biology, Czech Academy of Sciences, Květná 8, Brno, Czech Republic. [2] Department of Botany and Zoology, Faculty of Science, Kotlářská 2, Masaryk University, Brno, Czech Republic. [3] Department of Ecology and Vertebrate Zoology, University of Łódź, Łódź, Poland. ✉email: reichard@ivb.cz

Obligate brood parasites target a particularly valuable resource[1]—the reproductive effort of care-giving host species. By deceiving their hosts, they relegate the burden of brood care to the parents of another species[2], forcing them into costly alloparental care[3,4]. This outcome reduces, and often entirely eliminates, the benefits of parental care for the host's own offspring[5]. Negative consequences of brood parasitism for host fitness[2,6–8] can elicit the rapid evolution of host counter-adaptations[9,10]. The ensuing evolution of more refined parasite strategies to overcome host defences may turn into coevolutionary arms races between brood parasites and their hosts[11–13], sometimes leading to a host switch by brood parasites to escape host countermeasures[12]. Over recent decades, research on inter-specific brood parasitism in birds and social insects has considerably advanced our general understanding of coevolutionary dynamics[10,13].

Within this game played at the evolutionary scale, brood parasites and their hosts may hone their capabilities over individual lifespans. This response is primarily accomplished through learning—the alteration of behaviour (and the resulting behavioural outcome) obtained from repeated exposure to information[14]. The current evidence for learning in behavioural interactions between brood parasites and their hosts comes mainly from the host's perspective. There is evidence that hosts of avian and insect brood parasites use their past individual experience[15–17], as well as information shared through social learning[18–20], to enhance their behavioural responses to brood parasites. In contrast, there is little information on whether brood parasites are innately born with their complex behavioural repertoire or have to learn and refine their abilities over their lifetimes. This is especially relevant for generalist parasites that can learn from past experience to flexibly fine-tune their behavioural competence to the behavioural repertoire of particular host species. Indeed, most existing reports on parasite learning refer to broad generalist avian brood parasites[21–24]. The brown-headed cowbird (Molothrus ater) is capable of parasitizing over 200 host species[25] and can utilise social information acquired prior to their own parasitic attempts to choose suitable nests for parasitism[21]. Brown-headed cowbirds also have the cognitive abilities to remember promising hosts based on their past reproductive success[22] and improve their decision-making processes on whom to parasitise as they age[23]. Common cuckoos (Cuculus canorus) imprint on their natal habitat and return to similarly structured sites to reproduce[26,27], increasing their like-lihood of choosing suitable hosts[24]. However, these limited insights do not directly address the development of attributes that play a key role in successful brood parasitism – precisely timed and performed parasite oviposition followed by successful egg acceptance by the host. These are critical aspects of parasitism success as most host defence is concentrated on the period around parasite oviposition[13] and host defence may be costly to brood parasites[28]. Among birds, brood parasites often use stealth to lay their eggs[29] and individual brood parasites may be difficult to track. Hence, the role that learning plays in brood parasite success remains unclear.

We used the cuckoo catfish (Synodontis multipunctatus) to study how individual learning affects brood parasite success. Cuckoo catfish exploit the parental care of mouthbrooding cichlids in Lake Tanganyika by introducing their own non-mimetic eggs into host clutches[30]. Host spawning consists of two repeatedly alternating phases[31]. One includes courtship and defence of the spawning site from intruders. The other phase represents the actual egg laying; male and female circle around each other, one partner quivers its body and presents its flank while the other stimulates gamete release by nibbling near the genital papilla of its mate (termed male or female T-position

depending on which sex is stimulating gamete release by its partner)[32,33] (Supplementary Movie 1). Cuckoo catfish intrude an ongoing cichlid spawning, predate some cichlid eggs, and them-selves spawn[32] (Supplementary Movie 1). During these rapid encounters the catfish interrupt and disrupt the normal spawning sequence of the host cichlids, typically resulting in hasty egg collection by the host female when her discrimination of eggs is undermined, with the result that she collects catfish eggs with her own clutch[32]. The female cichlid then cares for the brood in her buccal cavity for approximately 3 weeks and releases independent offspring[34].

Mouthbrooding in fish is a demanding type of maternal care[34,35], comparable to the costly provisioning behaviour of avian parents. Similarly to some avian brood parasites[36], cuckoo catfish are highly virulent. The presence of cuckoo catfish eggs considerably reduces host clutches and often leads to their complete reproductive failure as parasite offspring consume host embryos[32,37]. Adult cuckoo catfish, in addition, are considered effective brood predators which feed on host eggs during host spawning[38,39]. Unlike most avian brood parasites (but see[21,40,41]), cuckoo catfish are readily amenable to laboratory-based experi-mental studies[32,37,39,42]. They reproduce year-round and their potential learning can be studied on a reasonable time scale.

Given strong interactions between host and parasites, indivi-dual learning of the hosts on how to cope with repeated cuckoo catfish intrusions over the experimental period would obscure detection of learned changes in parasite behaviour and their consequences. To decouple parasite learning from host learning, we used a riverine strain of Astatotilapia burtoni (from the Kalambo River, Zambia) as a host. A. burtoni inhabits Lake Tanganyika and surrounding water bodies, though our study population is unlikely to be exposed to cuckoo catfish parasitism in the wild. Mouthbrooding cichlids that are regularly exploited by cuckoo catfish in Lake Tanganyika[30] may have evolved innate recognition capabilities for the brood parasites, or an elevated aggression potential towards them. Further, sympatric hosts can learn to effectively eject parasite eggs during incubation[32]. In contrast, A. burtoni used in the study do not appear to learn to avoid catfish brood parasitism (Supplementary Table 1), similarly to evolutionarily naïve cichlid species from other African lakes[32]. A. burtoni do, however, present generalised aggression against any intruders to their spawning who may attempt to predate spawned eggs.

In this study, we test how repeated prior experience with host reproduction affected (1) the success of brood parasitism, (2) the costs incurred to the host cichlids and (3) the behavioural changes underlying the parasite learning process. We conducted experi-ments with 108 individual cuckoo catfish divided into three treatment groups that varied in their experience with host cichlids. The first group (naïve catfish treatment, $n = 36$) con-tained 5-year-old catfish, with no contact with cichlids prior to the experiment. The second group (experienced catfish treatment, $n = 36$) comprised of 5-year-old catfish housed with reproducing cichlid hosts for 1 year prior to the experiment. These catfish were raised using in vitro breeding and, therefore, were not imprinted on cichlid cues during incubation. The individuals from a third group (highly experienced catfish, $n = 36$) were kept with cichlids during their entire lifespans (i.e., for 7 years) and were raised naturally in their host´s buccal cavity, allowing potential imprinting on cichlid cues. Replicated groups of cuckoo catfish (3 pairs) and host cichlids (4 males, 12 females) were housed together in large experimental tanks (6 tanks per treat-ment) over a period of 4 months and host clutches were regularly inspected.

We demonstrate that prior experience of brood parasites with hosts improves their competence to overcome host defences.

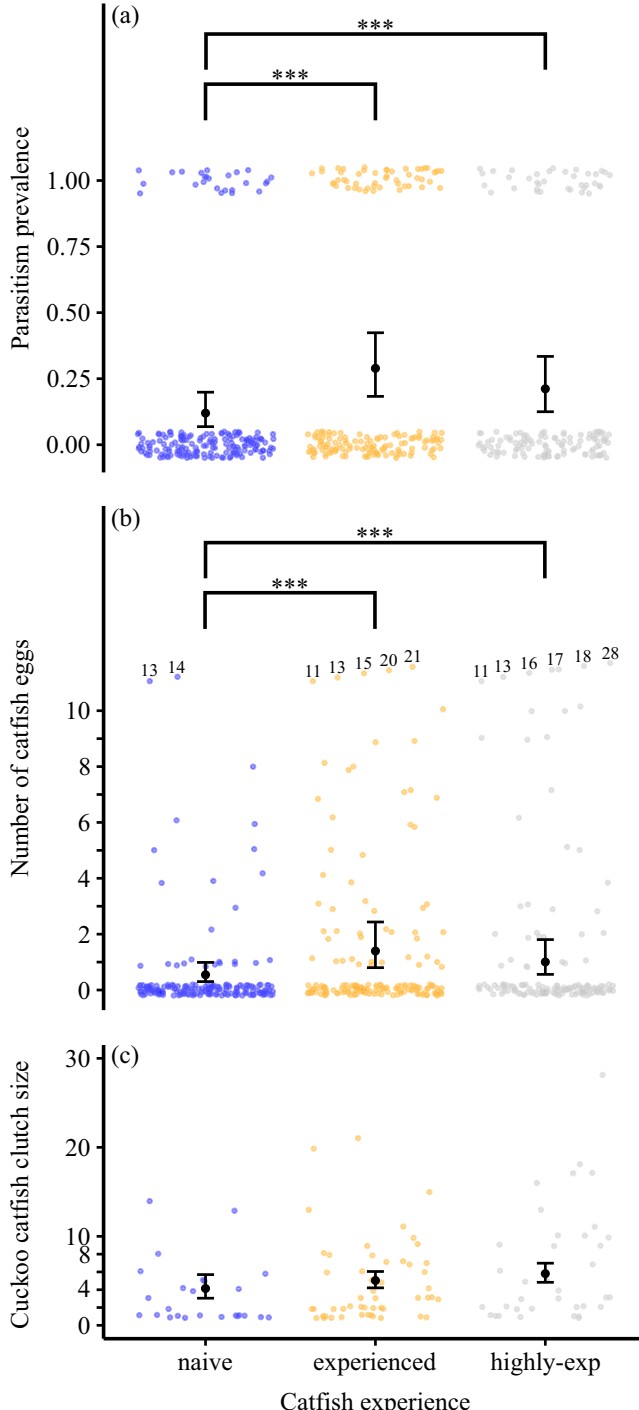

**Fig. 1 Naïve cuckoo catfish were less successful in parasitizing host clutches than experienced catfish.** Over the course of the experiment, (**a**) prevalence of parasitism and **b** overall number of parasite eggs in host clutches were significantly higher in experienced and highly experienced catfish compared to naïve catfish ((**a**): GLMM with binomial error distribution, $n = 509$ host clutches; naïve vs. experienced catfish: $z = 4.31$, $P = 1.63\text{e-}05$; naïve vs. highly experienced catfish: $z = 3.60$, $P = 0.000325$; (**b**): GLMM with negative binomial error; $n = 509$ host clutches; naïve vs. experienced catfish: $z = 4.17$, $P = 2.99\text{e-}05$; naïve vs. highly experienced catfish: $z = 3.63$, $P = 0.000282$), while (**c**) catfish clutch sizes did not differ significantly between the three treatments. The black dots represent the mean values of model predictions with their standard errors for the respective treatment (whiskers). Colour dots represent observed data points (jittered to improve their visibility) for naïve (blue), experienced (orange) and highly experienced (grey) catfish treatments. The experimental setup included 6 replicate tanks for each treatment. n for (**a**) and **b**: 191 host spawnings (naïve treatment), 174 host spawnings (experienced treatment), and 146 host spawnings (highly experienced treatment). n for (**c**): 25 parasitised host spawnings (naïve), 51 parasitised host spawnings (experienced), and 36 parasitised host spawnings (highly-experienced). Asterisks denote statistically significant differences (*** for $P < 0.001$). Numbers adjacent to data points in **b** represent values that were outliers (>10 eggs) and were lowered for graphical display to improve data visualization clarity. Source data are provided as a Source Data file.

## Results

**Parasite perspective**. Naïve cuckoo catfish had lower parasitism success compared to experienced and highly experienced catfish, suggesting they must learn to gain the experience to successfully parasitise their hosts (Fig. 1a). Overall, naïve catfish parasitised only 13% host clutches, while experienced and highly experienced catfish parasitised 25% and 29% host clutches, respectively. The lower prevalence of parasitism in the naïve catfish treatment was highly significant (Generalised Linear Mixed Model (GLMM) with binomial error distribution, $n = 509$ host clutches; naïve vs. experienced catfish: $z = 4.31$, $P < 0.001$; naïve vs. highly experienced catfish: $z = 3.60$, $P < 0.001$), but the probability of parasitising cichlid clutches did not differ between experienced and highly experienced catfish ($z = 0.75$, $P = 0.452$) (Supplementary Table 2).

The mean number of cuckoo catfish eggs calculated across all cichlid broods demonstrated the same pattern (Fig. 1b), with a mean ($\pm$ S.E.) of only 0.46 ($\pm$ 0.13) catfish eggs across host clutches in the naïve cuckoo catfish, but 1.42 ($\pm$ 0.25) catfish eggs and 1.63 ($\pm$ 0.35) catfish eggs in the experienced and highly experienced catfish, respectively. The lower number of catfish eggs in the naïve catfish treatment was significantly different from the experienced and highly experienced catfish (GLMM with negative binomial error, $n = 509$ host clutches: naïve vs. experienced catfish: $z = 4.17$, $P < 0.0001$; naïve vs. highly experienced catfish: $z = 3.63$, $P = 0.0003$, experienced vs. highly experienced catfish: $z = 0.55$, $P = 0.584$; Supplementary Table 3).

Once host defences were overcome, the number of parasitic eggs in host clutches was not affected by cuckoo catfish experience (Fig. 1c, Supplementary Fig. 1). The mean catfish clutch size (the number of catfish eggs found within the set of parasitised clutches) did not significantly differ between naïve, experienced and highly experienced catfish (mean $\pm$ S.E., naïve: 3.48 $\pm$ 0.73 eggs, experienced: 4.84 $\pm$ 0.66 eggs, highly experienced: 6.58 $\pm$ 1.07 eggs). No pairwise difference in the mean catfish clutch size was significant (GLMM with negative binomial error, host female body size and time since the start of the experiment as covariates: naïve vs experienced: $z = 0.04$, $P = 0.967$, naïve vs highly experienced $z = 0.01$, $P = 0.990$,

Overall, experienced and highly experienced cuckoo catfish had a two-fold higher parasitism success than naïve cuckoo catfish. Naive catfish steadily increased their success over the duration of the experiment and matched the parasitism rates of experienced catfish within 4 months. Behavioural quantification of cuckoo catfish intrusions of host spawnings demonstrated that higher parasitism success was achieved by better coordination and improved timing of the act of parasitism by the experienced cuckoo catfish. Parasite experience also increased direct costs for the hosts, irrespective of parasitism success, as the hosts associated with experienced and highly experienced catfish cared for smaller clutches, likely reduced through cuckoo catfish predation.

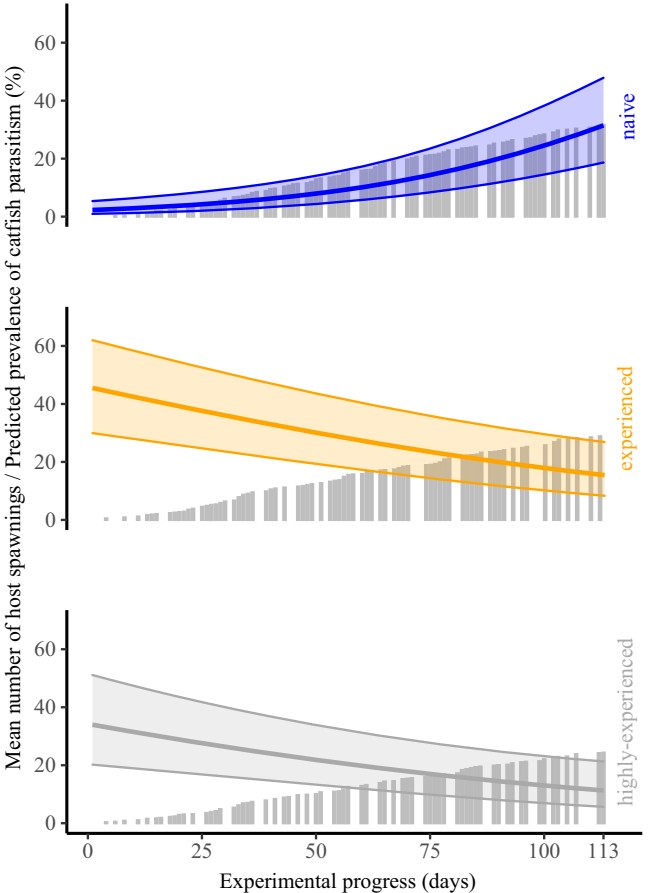

**Fig. 2 Relationship between the prevalence of brood parasitism by naïve cuckoo catfish and the number of host spawnings they witnessed.** Grey bars represent treatment-specific cumulative sums of host spawnings. The lines represent predicted probability of successful parasitism by cuckoo catfish for naïve (blue), experienced (orange), and highly experienced (grey) catfish based on a logistic GLMM (solid lines) and predicted standard errors for the predicted values (shaded areas). Source data are provided as a Source Data file.

experienced vs highly experienced $z = 0.05$, $P = 0.961$; Fig. 1c, Supplementary Table 4).

**Temporal trend in parasite learning**. In addition to clear differences between experimental catfish treatments, there was a rapid increase in brood parasitism success within the naïve catfish treatment over the 4-month experimental period. Prevalence (parasitism rate) in the naïve cuckoo catfish group increased from 3% during the first quarter of the experiment, to 25% in the last quarter (GLMM with binomial error: $n = 509$ host clutches, $z = 3.04$, $P = 0.002$; Fig. 2, Supplementary Table 2). The mean number of cuckoo catfish eggs likewise increased over time in the naïve catfish group, with a 20-fold increase between the first ($0.03 \pm 0.03$ catfish eggs) and the last ($0.63 \pm 0.21$ catfish eggs) quarters of the experiment (GLMM with negative binomial error: $z = 3.06$, $P = 0.002$; Supplementary Table 3, Supplementary Fig. 1). The difference between naïve catfish and more experienced catfish was most prominent at the start of the experiment and disappeared during the experimental period (GLMMs, interaction terms 'catfish experience: time progress'; for prevalence: naïve vs. experienced: $z = -3.78$, $P < 0.0002$, naïve vs. highly experienced: $z = -3.45$, $P < 0.0006$; for parasite eggs abundance: naïve vs. experienced: $z = -3.67$, $P < 0.0002$, naïve vs. highly experienced: $z = -3.50$, $P < 0.0005$). This contrast was

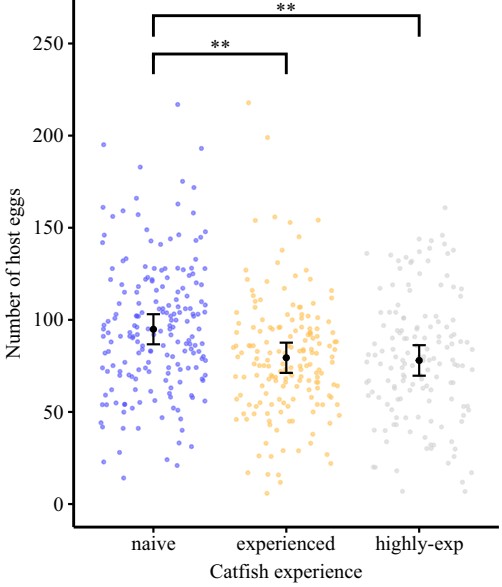

**Fig. 3 Host clutches were larger in treatments with naïve cuckoo catfish.** On average, host clutches in the experienced and highly experienced treatments were 17% smaller than the host clutches in the naïve treatment (GLMM with Gaussian error distribution; $n = 509$ clutches; naïve vs. experienced treatment: $z = -2.928$, $P = 0.00341$; naïve vs. highly experienced treatment: $z = -3.130$, $P = 0.00175$). The black dots represent mean values from model estimates and associated mean standard errors (whiskers) predicted across the experimental period. Coloured dots represent the observed data points (i.e., individual host clutch sizes) for naïve (blue, $n = 191$ spawnings in 6 replicate tanks), experienced (orange, $n = 174$ spawnings in 6 replicate tanks), and highly experienced (grey, $n = 146$ spawnings in 6 replicate tanks) cuckoo catfish treatments. Asterisks denote statistically significant differences (** for $P < 0.01$). Source data are provided as a Source Data file.

enhanced by a significant temporal decrease in the prevalence and mean abundance of parasite eggs in the experienced catfish treatment (and nonsignificant decrease in the highly experienced catfish treatment; Fig. 2, Supplementary Table 2, 3), but a rapid increase in parasite success in the naïve catfish was evident, irrespective of whether the comparison was made against other treatments (see above) or treatment-specific analyses compared against the null expectation of no temporal trend (Supplementary Table 5, 6).

Parasite clutch size did not vary significantly across the experiment in any catfish treatment (GLMM with negative binomial error: $z = 0.08$, $P = 0.937$; Fig. 3, Supplementary Table 4), indicating that cuckoo catfish must primarily learn to breach the frontline defence of their hosts, while their clutch size remains relatively stable.

**Host perspective**. Experienced (and highly experienced) cuckoo catfish decreased the size of host clutches. There were significantly more eggs in host clutches when they coexisted with naïve cuckoo catfish than in treatments with the two experienced groups (Generalized Linear Mixed Model (GLMM) with Gaussian error, naïve vs. experienced: $z = -2.93$, $P = 0.003$; naïve vs. highly experienced: $z = -3.13$, $P = 0.002$; Fig. 3, Supplementary Table 7), while there was no difference in host clutches between experienced and highly experienced catfish treatments ($z = -0.27$, $P = 0.787$). Host cichlid clutches contained $95.29 \pm 2.62$ eggs (mean $\pm$ S.E., $n = 191$ spawnings) when housed with naïve catfish, $79.18 \pm 2.40$ eggs ($n = 174$) with experienced catfish and $79.11 \pm 2.78$ eggs ($n = 146$) with highly experienced catfish. Because cuckoo catfish

raids on cichlid spawnings are, in addition to parasitism, associated with host egg predation[38], we propose that host clutch size differences reflect less effective host egg predation by naïve cuckoo catfish. Over the duration of the experiment, host clutch size (pooled across treatments) gradually decreased from $87.27 \pm 3.67$ eggs during the first quarter of the experiment to $82.89 \pm 3.77$ eggs during the last quarter. While this decrease was statistically significant overall ($z = -4.26$, $P < 0.001$, $n = 511$ clutches), the temporal trends did not differ among treatments (Supplementary Fig. 3, Supplementary Table 7).

**Behavioural mechanisms of parasite learning**. For a subset of interactions ($n = 9$ spawnings in the naïve catfish treatment, $n = 6$ in the experienced catfish treatment, and $n = 3$ in the highly experienced catfish treatment), we video-recorded spawning acts to understand how cuckoo catfish experience contributes to the successful exploitation of host cichlids. We concentrated on the timing of parasitism, since it plays a crucial role during cuckoo catfish intrusion of host spawning. As both parasites and hosts lay and fertilize their eggs externally, cuckoo catfish pairs must precisely time their oviposition and fertilization to coincide with those of their cichlid hosts. Cichlid T-positions (Supplementary Movie 1) are the most vulnerable moments during spawning because this behavioural sequence cannot be easily interrupted. Within the few seconds of a male adopting a T-position (i.e. when female lays her eggs), the cuckoo catfish have the best opportunity of successful egg predation and parasitism. Intruding as a group can make it more difficult for cichlids to defend against cuckoo catfish and, hence, social coordination of cuckoo catfish intrusions may additionally increase the chance of successful parasitism and predation.

Compared to naïve catfish, experienced and highly experienced catfish managed to intrude into more host spawning acts and demonstrated better timing and coordination. Naïve catfish missed five and four times more T-positions, respectively, than experienced and highly experienced catfish (19% vs. 4% and 5% for naïve, experienced, and highly experienced catfish, respectively; GLMM: naïve vs. experienced, $z = -3.87$, $P < 0.001$, naïve vs. highly experienced, $z = -3.67$, $P < 0.001$; $n = 18$ spawnings, Fig. 4a, Supplementary Table 8). During successful intrusions, it took naïve catfish on average 43% longer (mean $\pm$ S.E., $9.62 \pm 1.44$ s) to intrude on host spawning acts compared to experienced ($6.72 \pm 1.54$ s) catfish, and 17% longer compared to highly experienced ($8.25 \pm 2.86$ s) catfish, respectively (GLMM: naïve vs. experienced, $t = -3.35$, $P < 0.001$, naïve vs. highly experienced, $t = -3.79$, $P < 0.001$, $n = 18$ spawnings; Fig. 4b, Supplementary Table 9). Host male T-positions (where host oviposition occurs and which represents the most vulnerable moments during cichlid spawning) lasted for $5.52 \pm 0.12$ sec ($n = 115$ T-positions from 12 spawnings) and coincided closely with the reaction times of experienced catfish in most of their intrusions (horizontal dashed line in Fig. 4b). Naïve catfish also intruded host spawnings in smaller groups ($1.94 \pm 0.06$ individuals, $n = 227$ intrusions during 9 spawnings) compared to experienced cuckoo catfish ($2.71 \pm 0.08$ individuals, $n = 244$ intrusions during 6 spawnings; GLMM with Poisson error: $z = 3.60$, $P = 0.0003$) (Fig. 4c), but not compared with highly experienced catfish ($1.91 \pm 0.10$ individuals, $n = 113$ intrusions from 3 spawnings) (GLMM with Poisson error, naive vs. highly experienced: $z = 0.07$, $P = 0.946$; experienced vs. highly experienced, $z = -2.71$, $P = 0.007$) (Fig. 4c, Supplementary Table 10). There was no difference in host aggression towards intruding naïve, experienced or highly experienced cuckoo catfish (GLMM with negative binomial error: naïve vs. experienced, $z = 0.79$, $P = 0.428$, naïve vs. highly experienced, $z = 1.18$, $P = 0.238$), or

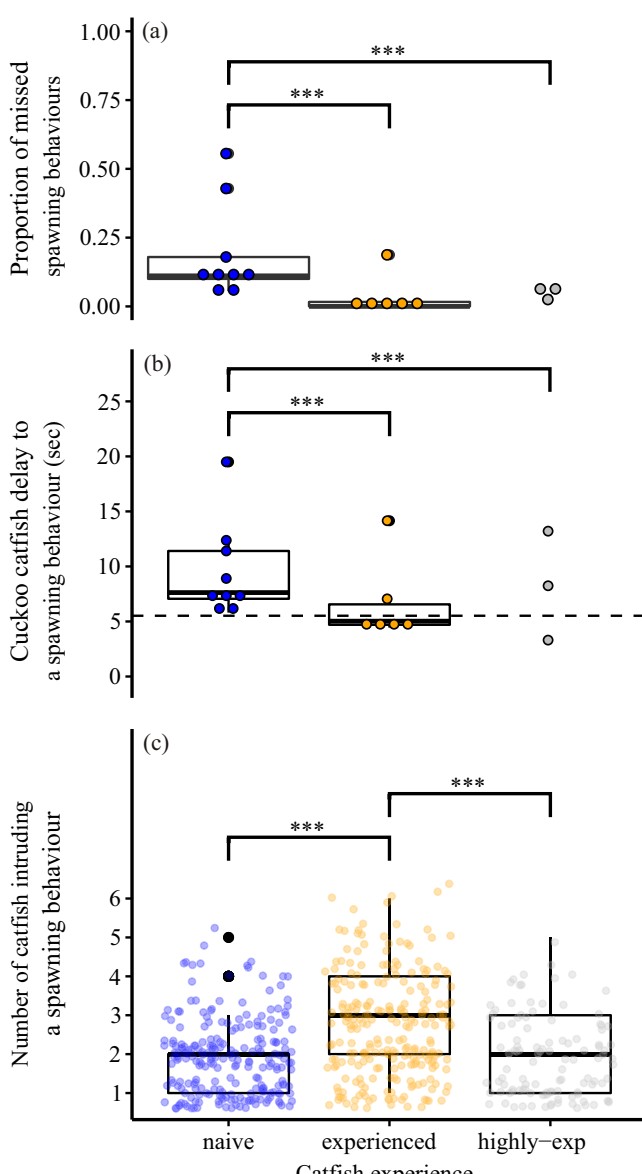

over the duration of the experiment (GLMM: $z = 0.54$, $P = 0.723$, $n = 18$ spawnings; Supplementary Table 11), supporting the observation that *A. burtoni* individuals do not learn to defend against intruding cuckoo catfish. We acknowledge the limited sample size of behavioural analyses, especially of the number of spawnings in the highly experienced treatment. Note that qualitatively the same results and interpretation were obtained from analyses when experienced and highly experienced treatments were pooled.

## Discussion

This study demonstrates experimentally that cuckoo catfish learn to overcome host defences, resulting in a significant increase in their efficiency as brood parasitises. This learning occurred during interactions with hosts during spawning, with the 4-month experimental period appearing to be sufficient time to alter behaviour. The higher parasitism success of experienced (and highly experienced) cuckoo catfish was primarily associated with more precise timing of the parasitic act. The hosts suffered an elevated cost of coexistence with experienced and highly experienced cuckoo catfish in terms of the decreased size of their own

**Fig. 4 Cuckoo catfish responses to cichlid spawning behaviour.** Catfish in the naïve treatment missed (i.e. did not intrude) a higher proportion of host spawning behaviours than their conspecifics in the experienced and highly experienced treatments (GLMM with binomial error distribution; naïve vs. experienced catfish: $z = -3.872$, $P = 0.000108$; naïve vs. highly experienced catfish: $z = -3.673$, $P = 0.000240$) (**a**). It took naïve catfish more time to react to (and intrude into) host spawning behaviours than experienced and highly experienced catfish (GLMM with Gamma error distribution; naïve vs. experienced catfish: $t = -3.348$, $P = 0.000814$; naïve vs. highly experienced catfish: $t = -3.787$, $P = 0.000152$) (**b**). The number of catfish intruding into host spawning behaviours was higher in experienced catfish groups than in the catfish groups of the other two treatments (GLMM with Poisson error distribution; experienced vs. naïve catfish: $z = -3.602$, P = 0.000316; experienced vs. highly experienced catfish: $z = -2.713$, $P = 0.006670$) (**c**). Dots represent observed data points for naïve (blue), experienced (orange), and highly experienced (grey) cuckoo catfish. Each data point represent one complete spawning of n independent host spawnings per treatment (**a**, **b**) or n intrusions to separated host spawning sequences (repetitive host T-positions) per treatment (**c**). n for (**a**) and (**b**): 9 host spawnings (naïve treatment), 6 host spawnings (experienced treatment), and 3 host spawnings (highly experienced treatment). n for (**c**): 227 intrusions (naïve), 244 intrusions (experienced), and 113 intrusions (highly experienced). Boxplots represent median (vertical line), interquartile range (IQR, box) and nonoutlier range (whiskers, 1.5 * IQR). Mean duration of cichlid T-position (most vulnerable to intrusion) is indicated by dashed line in **b**. Asterisks denote statistically significant differences (*** for $P < 0.001$). Source data are provided as a Source Data file.

clutch, irrespective of parasitism success, likely through increased cuckoo catfish predation on host clutches.

As the current study illustrates, experienced cuckoo catfish possess more competent behavioural responses to cichlid spawnings than naïve catfish. Hence, repeated exposure to host reproductive behaviour leads to an improvement of individual cuckoo catfish behaviours (e.g. more precise timing of oviposition), fitting the definition of successful learning[14] and generating improved parasitism success. It has been shown previously that learning is an important component of evolutionary processes[43], and learning and natural selection are tightly linked in animal populations[44–46]. Greater cognitive abilities, as well as enhanced problem-solving capabilities, confer higher reproductive success[47,48]. Among brood parasites, female cowbirds have enhanced spatial memory performance compared to nonparasitic birds as well as compared to male cowbirds[49]. Female cowbirds recognise which characteristics of the nest, the host, and nestlings in the nest, including their temporal changes, provide higher chances for successful parasitism. They use this information to choose and parasitise the most suitable nests to maximize their reproductive success[22]. Among fishes, superior spatial learning is positively associated with the reproductive success of non-territorial male bitterling (*Rhodeus amarus*). Fertilization of bitterling eggs occurs in living freshwater mussels, and males with the most effective sperm ejaculation strategy (in terms of both spatial and temporal decisions) gain higher shares of paternity[50]. Likewise, naïve cuckoo catfish appear able to evaluate host spawning situations and adjust their behavioural responses accordingly. This effect, in addition to other indirect evidence[23], underpins the importance of learning for reproductive success of brood parasites and suggests that brood parasites are not necessarily innately effective in their reproduction. The rapid increase in parasitism success of naïve catfish during our experiment indicates that learning occurs relatively quickly and cuckoo catfish can become fully competent to parasitise their hosts within

4 months of continuous exposure to hosts and after experiencing approximately 30 host spawnings (Fig. 2a). Similarly rapid learning was previously detected in avian hosts[18,51]. The ability for both hosts and brood parasites to adjust their behavioural responses to new conditions indicates highly dynamic behavioural interactions within brood parasitic systems, with the potential to allow one partner to overcome an evolutionary lag[52] or to accelerate co-evolution[46].

The two-fold higher parasitism success of experienced and highly experienced catfish compared to naïve cuckoo catfish arose from a higher number of successfully parasitised clutches, but not from a higher number of parasitic eggs in individual clutches. Up to 14 cuckoo catfish embryos were recovered from cuckoo catfish hosts collected in the wild[30] which is comparable to the numbers we typically observed in our experiment (Fig. 1c). More than one cuckoo catfish offspring often successfully develop in a parasitised host clutch[30,32,37], a situation also observed in some nonevicting avian brood parasites[53], where multiple parasitic eggs and chicks may coexist in the same host nests and are able to fledge successfully[54]. The lack of a difference between naïve and more experienced cuckoo catfish in their clutch sizes suggests that, once spawning occurred, egg acceptance by the host female was not affected by cuckoo catfish experience. This outcome appears comparable to avian brood parasites where, once the host nest has been successfully parasitised, parasites have a limited capacity to prevent egg rejection by the host (but see[55]).

Collectively, our findings suggest that cuckoo catfish gain experience on how to successfully parasitise their hosts. Enhanced efficiency through learned experience from past parasitism attempts may equip cuckoo catfish with the abilities needed to successfully reproduce. Beside an evolutionary arms-race, there appears an equally relevant race taking place at the level of individual behaviour over much shorter time spans. From avian brood parasitic systems we have ascertained that host individuals can limit parasitism following their past individual experience[17,56] or socially distributed information[18,20], but learning in parasites was less obvious because the behaviour is typically stealthy. The capacity to quickly adapt may be a key aptitude for successful brood parasite reproduction in the face of continuous learned and evolved improvements in host defences. Alternatively, the ability to fine-tune their parasitic behaviours may enable generalist brood parasites to utilize a broad range of potential hosts and thus escape a possible evolutionary trap in an escalated coevolutionary arms-race with one host species, as suggested for avian brood parasites[12]. Using the cuckoo catfish, we demonstrate that brood parasites learn to improve their abilities to successfully parasitise their hosts. This finding illustrates the complexity of an ongoing struggle between brood parasites and their hosts, taking place at both individual and evolutionary levels.

## Methods

**Study system.** The cuckoo catfish (*Synodontis multipunctatus*) belongs to the African catfish family Mochokidae. The genus *Synodontis*, with 131 species distributed across African freshwaters[57], gave rise to a small radiation in Lake Tanganyika, with 10 described endemic species[58]. The taxonomy of the group is not well established[59] and we use the name *S. multipunctatus* as this species is confirmed as a brood parasite[30] and the name was used in previous studies[4,30,32,37,42]. Cuckoo catfish primarily parasitise mouthbrooding cichlids from the tribe Tropheini[30], but species from other lineages can also be parasitised[59].

**Experimental design.** All experiments took place between January and August 2020 at the Institute of Vertebrate Biology, Czech Republic. Prior to experimental use, fish were housed in mixed-sex groups in tanks equipped with shelter and internal filtration. Cuckoo catfish were F1 generation of commercially imported wild-caught parents (10 pairs). Host cichlids were descendant of wild fish imported from Kalambo, Zambia. Experimental tanks (420 L; length 150 cm, depth 70 cm, height 40 cm) were equipped with internal filtration, fine gravel (2–4 mm

diameter), half a clay pot as a shelter on each side of the tank, and one artificial plant in the centre of each tank. Water temperature was maintained at 27 °C (±1 °C) and the dark - light regime was set to 11 h:13 h. In total, we stocked 18 tanks with 4 males and 12 females of the mouthbrooding cichlid *Astatotilapia burtoni* and introduced 3 cuckoo catfish pairs of one of three different experience levels. Naïve catfish (*n* = 36 individuals) had no prior experience with cichlids. Experienced catfish (*n* = 36) were housed together with reproductive cichlids for 12 months prior to the experiment and were age-matched to naïve catfish (5 years old). Highly experienced catfish (*n* = 36) were raised, coexisted and reproduced with cichlids for 7 years (and were on average 7–8% larger than both naïve and experienced catfish; mean ± SE, naïve: 116.2 ± 1.9 mm, experienced: 117.1 ± 1.5 mm, highly experienced: 125.6 ± 1.4 mm; Linear Model (LM): experienced vs. highly experienced, estimate ± S.E = 8.44 ± 2.29, *t* = 3.68, *P* = 0.0004, experienced vs. naïve, estimate ± S.E = −0.94 ± 2.29, *t* = −0.41, *P* = 0.681, *n* = 108). Additionally, both naïve and experienced cuckoo catfish were bred using in-vitro fertilisation[32] to avoid cichlid imprinting (i.e., priming with cichlid cues), while highly experienced catfish were bred under natural conditions within the buccal cavities of their hosts. Each experimental tank contained catfish with the same experience level. Due to space limitations, we split the experiment into two consecutive phases with 3 replicate tanks for each treatment within both phases (in total 9 experimental tanks per phase). Between the two experimental phases, host cichlids were placed together and haphazardly assigned to new experimental tanks. During the second phase, we removed some cichlids from the tanks because of injuries caused by their intraspecific aggression (3 males and 3 females in total), and those hosts were not replaced. Over an experimental phase, cuckoo catfish and cichlids freely interacted for 15–16 weeks. During this period, each tank was checked for mouthbrooding cichlids twice each week (Tuesday and Friday). We caught the mouthbrooding females, gently washed the eggs out of their mouths using a jet of water from a Pasteur pipette, measured their body size to the nearest mm, and released them back to their experimental tank. For each female, we counted the number of cichlid eggs and cuckoo catfish eggs (if present). At the end of each experimental phase, we measured body size of all cuckoo catfish to the nearest mm. There was no significant difference between the number of cichlid spawnings between naïve and experienced catfish treatments (Generalised Linear Models with negative binomial error distribution, estimate ± S.E.: −0.093 ± 0.145, *z* = −0.644, *P* = 0.519), nor between naïve and highly experienced catfish (estimate ± S.E.: −0.269 ± 0.148, *z* = −1.810, *P* = 0.070).

**Behavioural recording**. Over the experimental period, we successfully recorded 18 videos of spawning events (Lamax x3.1 ATLAS cameras; naïve catfish treatment, *n* = 9; experienced catfish treatment, *n* = 6; highly experienced catfish treatment, *n* = 3). One camera was placed near the spawning site approximately 20 cm away from spawning activity and a second camera was placed outside the experimental tank to obtain an overall view. Nine spawnings were recorded from the start (*n* = 7 naïve catfish experiments and 2 experienced catfish experiments) and nine spawnings were recorded from the timepoint when we recognised ongoing spawning activity (*n* = 2 naïve, 4 experienced, and 3 highly experienced catfish experiments). From the video footage taken for each spawning, we scored all overt aggression that host cichlids directed towards cuckoo catfish, counted the number of intruding catfish during each distinct cichlid spawning behaviour (i.e., male and female cichlid interact in a repeated succession of quivering and T-positions), measured the delay of intruding catfish to each distinct spawning behaviour (i.e., the time from the start of spawning behaviour until the first catfish directly approaches the spawning cichlids), and recorded the presence or absence of catfish during each spawning behaviour. Additionally, we recorded whether cichlids used the available shelters for spawning as this might have impeded catfish recognition of the spawning activity. When spawning was recorded from the start, scoring started 100 s before we detected the first egg laid (cichlid or cuckoo catfish). When spawning was already ongoing, the scoring started immediately after the cameras were in place. Mounting of the cameras did not interrupt the normal behaviour of cichlids or catfish. For all video footage, scoring ended 100 s after the last male-female interaction within the spawning site. To estimate the duration of male T-positions during spawnings, we measured the time period from the start of male nuzzling near female genital papilla until the female turned around either to collect eggs or start nuzzling near the male´s genital papilla (*n* = 115 male T-positions from 12 cichlid spawnings).

**Statistical analysis**. We used R v. 3.5.1 (R Development Core Team, 2018) for all statistical analyses. All statistical tests were two-sided. First, we compared body size among the three cuckoo catfish experience levels using a Linear Model with catfish size (mm) as response variable and 'treatment' (naïve, experienced, and highly experienced catfish) as predictor variable. Second, we formally tested whether the number of host spawnings varied between the treatment groups (total numbers: naïve = 191, experienced = 174 spawnings, highly experienced = 146 spawnings). To obtain an insight into temporal dynamics of cichlid spawning, we calculated the number of cichlid spawnings for each treatment in each quarter of the duration of the experimental period. We fitted a GLM with a negative binomial error distribution (to account for slightly overdispersed data) with the number of cichlid spawnings as the response variable and our treatment groups as predictors.

To test how experience with host spawning (treatment) affected cuckoo catfish ability to place their eggs in the care of the host, we compared (1) the number of parasitised cichlid clutches among the three catfish experience groups (prevalence of parasitism), (2) the mean number of catfish eggs introduced into cichlid clutches among the three treatment levels (mean parasite egg abundance, the mean number of catfish eggs calculated across all cichlid broods), (3) mean parasite clutch size (the number of catfish eggs calculated only across parasitised cichlid broods), and examined (4) temporal dynamics of all three measures of parasite success within each treatment group throughout the duration of the experiment.

To test for differences in prevalence of parasitism among different cuckoo catfish experience treatments, we applied a Generalised Linear Mixed-effects Model (GLMM, R package *glmmTMB*)[60] with a binomial error distribution. We fitted the occurrence of 'catfish parasitism' (1 = yes, 0 = no) as the binary response variable and 'treatment effect' (i.e., 'catfish experience'), 'time progress of experiment' (1–113 days) and 'host female body size' (in mm) as predictor variables. We additionally fitted an interaction between treatment ('catfish experience') and 'time progress of experiment' to the model to test whether parasitism rate changed over time at treatment-specific rates. We included tank identity ('tank ID') as a random intercept to account for nonindependence of data obtained from the same tank.

Next, we tested whether the mean number of parasite eggs that were accepted by host females during one spawning bout differed between catfish experience treatments. We applied two GLMMs (R package *glmmTMB*)[60] with a negative binomial error distribution (i.e., *nbinom1*) to account for over-dispersed count data. We applied GLMMs on the mean abundance of catfish eggs (across all host clutches) and on mean clutch size of cuckoo catfish using a subset of clutches that were parasitised. For both GLMMs, we included the 'number of cuckoo catfish eggs per clutch' as the response variable and treatment ('catfish experience'), 'time progress of experiment', and their interaction as predictor variables. We additionally fitted 'host female body size' as a predictor variable because larger female cichlids are capable of laying more eggs and may appear more attractive hosts to cuckoo catfish. Further, a higher number of host eggs may increase the number of opportunities for cuckoo catfish to deposit their own eggs in the host clutch. 'Tank ID' was included as random intercept to account for nonindependence of data.

To test whether cuckoo catfish presence affected cichlid spawning activity, we applied a GLMM (R package *glmmTMB*)[60] with Gaussian error distribution (which provided superior model fit compared to Poisson and negative binomial distributions by 'simulateResiduals' and 'testDispersion' functions in the R package *DHARMa*)[61]. We fitted the 'number of host eggs' per clutch as the response variable and treatment ('catfish experience'), 'host female body size', 'time progress of experiment', and 'experimental phase' (1st or 2nd phase) as predictor variables. We also included 'tank ID' as random intercept to account for nonindependence of data. The full model further included an interaction between treatment and 'time progress of experiment' to accommodate the possibility that host egg numbers may be affected differently across catfish experience treatments over time. As this full model predicted no difference in temporal aspect of host clutch size among treatments ('catfish experience': 'time progress', experienced: z = 0.92, P = 0.360, highly experienced: z = 1.46, P = 0.143), we subsequently dropped the interaction term from the final model.

We used data collected from video footage to investigate whether naïve, experienced and highly experienced cuckoo catfish differed in their response to host spawnings and, additionally, if catfish from the three treatments elicited different host reactions towards them by applying Linear Mixed-effect Models using the R packages *lme4*[62] and *glmmTMB*[60]. To account for different starting times of recordings, we calculated either the rate of behaviour per minute of observation (i.e., for aggression) or their relative values (i.e., for the number of host courtships that cuckoo catfish missed).

First, we tested whether host spawning pairs increased their aggressions towards cuckoo catfish over the experimental period to rule out the presence of host adaptation to cuckoo catfish intrusions, which would interfere with our aim of understanding parasite learning. We fitted a Generalised Linear Mixed-effects Model (GLMM, R package *glmmTMB*) with a negative binomial error distribution. The number of overt aggressive behaviours that the spawning pair performed towards cuckoo catfish per minute of catfish presence at the spawning site (summed over male and female cichlid) was fitted as the response variable and treatment ('catfish experience') as the predictor variable. We further included 'time progress of experiment' and 'experimental phase' as predictors to account for their possible effect on host aggression. We additionally included 'tank ID' as random intercept in the model to account for individual variation in host aggression levels among experimental tanks.

To investigate if naïve cuckoo catfish missed more opportunities to parasitise cichlids than experienced and highly experienced catfish, we fitted a GLMM (R package *lme4*) with a binomial error distribution. We included the proportion of missed spawning behaviours (coded as 'missed spawnings behaviours' versus 'intruded spawning behaviours', based on count data for each spawning) as the response variable ('spawnings missed') and treatment ('catfish experience') as a predictor variable. We fitted 'tank ID' as a random intercept to the model to account for nonindependence of data within tanks, and we additionally fitted a random intercept based on whether the spawning was covered by a shelter or not ('sheltered spawn', yes / no) since spawning in a shelter may have been less apparent to catfish.

We tested whether cuckoo catfish experience played a role in the timing of their intrusion to specific spawning behaviours by fitting a GLMM (R package *lme4*) with a Gamma error distribution to account for a positive skew in the data distribution. We included the mean delay of catfish to the first appearance of cichlid T-position in seconds ('catfish delay', see main text and Supplementary Movie 1 for a detailed description of cichlid spawning sequence) as the response variable and 'catfish experience' as the predictor variable. We included 'tank ID' and 'sheltered spawn' as random intercepts.

Finally, we fitted a GLMM with a Poisson error distribution to test whether cuckoo catfish learn to synchronise their intrusion behaviour as they gain experience through interactions with their hosts. We included the maximum number of catfish during a specific cichlid spawning behaviour ('intruder number', count data) as the response variable and 'catfish experience' as the predictor variable. To account for nonindependence of data within experimental tanks and spawnings, we included a random intercept where each spawning was nested within 'tank ID' in the model.

**Ethical compliance**. Research adhered to all national and institutional animal care and use guidelines, was administered under permit No. CZ62760203 and was approved by ethical boards of the Institute of Vertebrate Biology and the Czech Academy of Sciences (approval No. 32-2019).

**Reporting summary**. Further information on research design is available in the Nature Research Reporting Summary linked to this article.

## Data availability
The data generated in this study have been deposited in the Figshare repository accessible via (https://doi.org/10.6084/m9.figshare.14822838.v4), including all source data for figures. Source data are provided with this paper.

## Code availability
Code to reproduce the analyses and figures of this study are stored in the public repository (https://doi.org/10.6084/m9.figshare.14822838.v4).

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

## Acknowledgements

We thank Carl Smith, Kristina Sefc, Richard Bailey and Marcel Honza for comments on the manuscript. Funding came from the Czech Science Foundation, grant 18-00682 S (to M. R.). Part of the research was funded by the Austrian Science Fund (FWF) (Grant number J4584-B to H. Z.). For the purpose of Open Access, the authors have applied a CC BY public copyright licence to any Author Accepted Manuscript (AAM) version arising from this submission.

## Author contributions

H.Z., R.B., M.P., and M.R. conceived and conceptualized the study; H.Z., R.B., M.P. collected data; M.R. acquired funding and supervised the study; H.Z. analyzed data; H.Z. and M.R. drafted the manuscript; all authors edited manuscript and approved submission.

## Competing interests

The authors declare no competing interests.
