## [Peer Review File · Nature Communications]

Individual experience as a key to success for the cuckoo catfish brood parasitismReviewers' Comments:

Reviewer #1:

Remarks to the Author:

This is an important study that uses 3 powerful treatment groups to assess the role of experience in two different ways in the effectiveness of brood parasitism by parasitic catfish in captivity. I believe that there are preceding papers on this by David White on captive female cowbirds of different ages (i.e. age-specific parasitism patterns) and by Michael Taborsky (i.e. on habitat choice by captive female cuckoos), which must be cited by this study, too.

Line 24: I don't understand how there can be an arms-race ontogenetically? Please explain better.

Line 41: cite D White's works here on captive female cowbirds.

Line 44: please define learning (e.g. Barron et al. 2015 Trends in Neurosciences)

Line 54: but see Christi Riehl's work on greater ani's and personal experience guiding parasitism (intraspecific) decisions? Is that ontogeny/learning?

Line 67: start new paragraph here

Line 73: I wouldn't say this (cite Dave White)

Line 81: very nice experimental set up--you're able to test age vs. experience!!

Line 114: if juvenile catfish feed on host eggs, then why do the adults do it too? Interesting question to me.

Line 149: post hoc tests between 1 vs. 7 year experimental groups, too? Should be reporting on those, too.

Line 174: actually the host was defenceless in this experiment (since you used burtoni, which is not coevolved with the catfish). Please rephrase this sentence.

Line 199: naked p value. You always need to show the test-statistics, please, not just the p value.

Line 297: why not use a Mann-Whitney test here?

Line 301: what if you didn't combine it and used n=6 for one of the groups?

Line 331: again, need a definition for learning vs. experience.

Figs 1-2. there is something very wrong with the color scheme--the naive group should be in blue but it's indicated as red???

Fig. 3--please show, analyze, and discuss these same patterns for the two experienced treatment groups, too. Essential!

Fig. 4: please indicate significant differences with a "*".

Reviewer #2:

Remarks to the Author:

This study demonstrates that brood parasitic catfish that have more experience in reproducing with cichlid hosts have higher reproductive success than host-naïve catfish. The study is a good read and the methods are sound. The results are clear and interesting, although one has to admit that they are also not particularly surprising given that higher experience with reproduction enhances reproductive success and efficiency in many species. I have a number of specific points and questions below.

Line 44: I do not entirely agree. Imprinting is also a form of learning, and here we know at least that it happens, and that it does in a specialized brood parasite. The learning of habitat features early in life (habitat imprinting) has been shown in an avian brood parasite (Kolecek et al. 2020, *Science of Nature*; Teuschl et al. 1998, *Anim Behav.*)

If *A. burtoni* were naïve and had no adaptations to fend off parasites. Are *A. burtoni* ever used as hosts in nature? And, while I can understand the reasons for using naïve hosts, the question is how generalizable the results are for natural systems, in which hosts will have experience with catfish. How relevant is the learning of social cues or timing by catfish for their success in naturally parasitized systems ?

As the methods come afterwards, it would help to have some basic information here:

-line 82: How many catfish in each treatment?

-Line 86/87: How many male and female catfish and cichlids per tank, and how many tanks per treatment? Up to this point one may think there was only one tank per treatment, but later it becomes clear there were 6 of them

line 88: 'strong reciprocity' is a term used in the cooperation/altruism literature, so please rephrase to avoid misunderstandings

line 117: was predation confirmed by video analysis? Could there be alternative explanations or reasons why the cichlids had different clutch sizes?

How was the number of catfish eggs corrected for number of host spawning events – see also my question below about the number of cichlid spawnings in the treatments. If that differed systematically between treatments, part of the results may not be explained by catfish learning but by the chances of catfish to disseminate their available eggs. If, as an extreme example, in one treatment twice as many spawnings occurred than in another, but catfish have only a certain amount of mature eggs to spend, then the proportion of parasitized clutches, the number of eggs across cichlid clutches (and possibly even the catfish clutch size) would be affected.

In addition to Fig. 1 the raw data points should be plotted to see how the interaction comes about, and whether the data are equally distributed along the x -axes, or whether there are time patterns.

Even if not significant, the success of the experienced catfish declines over the course of the experiment, which in part drives the significant interaction. How does this come about, if *A. burtoni* hosts really do not learn?

Please correct the in-figure labels of Fig. 2, which say that blue is highly experienced. I assume the legend is correct though, which claims that blue is naïve, and so it fits the results.

Fig 4: should be separated by experienced and highly experienced to avoid the argument that the differences are merely because of the higher age of the highly experienced catfish.

line 134: sample size of subset?

line 146: explain "T-position" for non-cichlid readers

line 167: check reference numbers. ref 26, for instance is "26. Materials and methods are available in the supplementary materials." which does not seem to fit here

line 172: is there any evidence for evolutionary traps in this system?

In the methods, please give the exact number of host spawnings in each treatment, and please test if there was an effect of treatment on number of cichlid spawning

line 269: so what happens when cichlid egg number is taken in the models instead of the proxy of cichlid female size?

line 274: Cichlid spawning activity includes number of spawnings, not only number of eggs. See above – are there treatment differences?

Reviewer #3:

Remarks to the Author:

Review of: Learning in brood parasites: experience as a key to success for the cuckoo catfish

While there is a fair bit of work in the avian brood parasitism literature on the role that information and learning plays in refining host defences, understanding the role that learning plays in refining the behavior of brood parasites is something that has basically gone unstudied, because the systems are just not well suited for this kind of work. The authors here have used the practical and logistical advantages of their unique fish system to experimentally investigate how learning refines parasitic behaviours, which is great and in my opinion worthy of serious consideration for publication in Nature Communications.

That being said, the execution of the manuscript is (frankly) pretty ordinary, which made this a very interesting paper to review. Overall, given that the actual study is very novel and the execution of the experiments is suitable, the paper should be given a chance for resubmission, as these results should be published in a top-tier journal. Of course, that can't happen if the manuscript isn't good enough, so a very serious overhaul is needed.

In particular, my main issue(s) all seem to stem from the same key issue: the frivolous use of language and the generally uninformed pitch of the paper seems to stem from a (seemingly very) superficial understanding of the non-catfish related brood parasitism (and broader coevolution) literature. I think the most upfront evidence of this is the stark contrast between the thoroughness of the referencing and detailed writing with which the authors discussed catfish literature (e.g. paragraph starting L 55) compared to the previous paragraphs of the introduction, and there was an equally superficial discussion of the literature and how the results fell within it and added to it in the results/discussion section. I have given some examples below, but generally a much more thorough knowledge of the relevant literature (i.e. you're pitching this paper as a general coevolution/brood parasitism paper – you need to know this literature) and a more thoughtful placement of this result within this context (including discussion) is needed.

I sincerely think that this result is suitable for publication in Nature Communications, and hope that the manuscript is given an opportunity for resubmission.

General writing:

Abstract

L 14-15: It's true that evidence (especially direct evidence) of learning in brood parasites is very rare,

but saying the role of "other factors... is poorly understood" is untrue. There is a lot of literature on the role that social and personal information have on host behaviours, at least in birds, as well as the effects other factors have in shaping these interactions, so I would tone down this kind of language.

L18-19: This is not true. Presumably you mean interspecific brood parasitism (brood parasitism is too vague on its own for a statement like this – probably also applies to L13), but even so, interspecific brood parasitism has recently been described in coral reef fishes (e.g. Tarel et al. 2020 *Molecular Ecology* <https://doi.org/10.1111/mec.15243>)

L 23: Please change "superior" to "improved" (or something like this).

L 23-25: The language here does not sit well with me – this is not the first study to demonstrate that an arms race occurs throughout the lifespan of brood parasites and hosts, but you probably (to my knowledge) could argue that you are providing the first direct evidence (or at least among the first) of learning in brood parasites with regards to their interactions with hosts.

Introduction

L 26-27: I don't really see the point of this first sentence – parasitism is successful because parasites are abundant? In my opinion that's not really an opening sentence for a journal like *Nature Communications*. Generally, this problem (lots of words not saying much) is pretty ubiquitous in the introduction – more thorough incorporation of the literature is needed.

L 27-28: "Particularly costly" compared to what, and if so, why isn't it referenced? This seems a very frivolous or careless use of language and is something needs to be addressed.

L 29: "rapid evolution of host counter-adaptations" should be referenced (maybe Spottiswoode & Stevens 2012 *Am Nat* would be an appropriate reference here).

L 34: Is the word "rapid" really needed? I understand it makes it sound more urgent, maybe, but I think "coevolutionary dynamics" is fine, and more accurate. This should also be referenced (maybe a review or two – there are plenty of great ones to choose from).

L 36-37: This is not true, and disregards a very substantial literature on the topic, at least in birds (see work by Campobello, Davies, Feeney, Langmore, and references within their papers).

L 41: I do not think that the terms "parasitic skills" and "master their abilities" are appropriate language for a journal like this.

L 43-45: Where are the references to support this statement? There is also a lot of information on the life history traits that brood parasites (not just cowbirds) target, which is relevant here.

L 53: This is not always true, see work by Glog and/or Soler. Also probably looking up mafia hypothesis work.

Paragraph starting L 55: This is the first paragraph that I have felt is of the quality expected for a journal like *Nature Communications*. The author should note how superficial and sparsely referenced the previous paragraphs are compared to this. It appears as though, while they certainly know their stuff regarding this system, they are less aware of the other relevant literature on the topic, which is very clear in their writing. I suggest a major overhaul of the majority of the introduction, and a more thoughtful use of language and care used when developing the context for this work.

Results & discussion

It is very difficult to review the stats when they are (sometimes) in the figure captions. I am not sure

if Nature Comms accepts this format, but for readability, I find it much more helpful having the stats in the manuscript text.

L 101: Please change "superior parasitic skills" to something more appropriate.

L 102: Please try and get rid of the word "skills" its vague and not very helpful. It also just sounds a bit silly.

L 103-104: Percentages are not particular informative here by themselves and I'd suggest providing N wherever possible as well. Also, you should be providing more information with your stats (e.g. χ^2 and df with your glmm, or whatever depending on the model). I would also suggest providing an R Markdown so that we can check the stats more thoroughly.

L114: Parasite experience, or experience with a parasite?

L 130-133: It's worth noting that other studies in birds (albeit on hosts) have found that learning can also be extremely quick (Langmore et al. 2012 Behav Ecol, Feeney & Langmore 2013 Bio Lett), and this study is interesting because you're finding similar evidence in the parasite (which is something we just can't really do in birds).

L139: The results are just very inconsistently presented. Here, the stats are presented differently and more correctly than they are above (L104) – why? Also, why are some stats presented in figure caption, but others are in text? It just seems like the paper is a bit of an unpolished draft, rather than a polished manuscript ready for submission to a major journal, which is a bit irritating, because your result is so interesting.

L 143-151: You should probably include references to supplementary videos and/or screenshots throughout this paragraph.

Paragraph starting L 161: This is way too superficial (as is the rest of the discussion parts of this results/discussion section). Much more insightful and thorough consideration of the brood parasitism literature is required, how this study fits in, and the broader implications of this result. This requires substantial work, as it is currently no where near the quality I would expect for a journal like this. That being said I think you can fix it, but I would almost suggest reaching out to someone who works on this to help, as it currently falls (very) short.

L 168: I mean, your fish system is just more appropriate for examining this, because you can put them in tanks. You should just say this, because its true.

Methods

Generally seem fine and well written. I would strongly suggest submitting an R Markdown file with all analyses so that we can more easily assess the suitability and execution of the stats.

REVIEWER COMMENTS

Reviewer #1 (Remarks to the Author):

This is an important study that uses 3 powerful treatment groups to assess the role of experience in two different ways in the effectiveness of brood parasitism by parasitic catfish in captivity. I believe that there are preceding papers on this by David White on captive female cowbirds of different ages (i.e. age-specific parasitism patterns) and by Michael Taborsky (i.e. on habitat choice by captive female cuckoos), which must be cited by this study, too.

RESPONSE: We now refer to the work by D. White (which was also cited in the previous version) more often (White et al. 2007, 2017, White 2020) and also refer to work by M. Taborsky's team (Teuschl et al. 1998, Vogl et al. 2002).

Line 24: I don't understand how there can be an arms-race ontogenetically? Please explain better.

RESPONSE: We have reworded the abstract and incorporated this point into the altered text.

Line 41: cite D White's works here on captive female cowbirds.

RESPONSE: We have moved our reference to D. White's work earlier in the paragraph (we did cite it later in the previous version). We also toned down the statements on the non-availability of information on brood parasites learning abilities. Finally, we added another reference to a study by D. White in this paragraph (please see lines 41-42, 46).

Line 44: please define learning (e.g. Barron et al. 2015 Trends in Neurosciences)

RESPONSE: We would like to thank the referee for flagging this very helpful conceptual paper (cited now on lines 35-37). Accordingly, we now clearly define learning in the context of our experiment and how it is achieved by cuckoo catfish (lines 245-248).

Line 54: but see Christi Riehl's work on greater ani's and personal experience guiding parasitism (intraspecific) decisions? Is that ontogeny/learning?

RESPONSE: We reworded "notoriously" difficult to follow for "may be" difficult to follow (line 58). We believe that reference to cooperative breeding and associated intra-specific parasitism here would be misleading and create a potential for misunderstanding (albeit we agree that the decision process is similar).

Line 67: start new paragraph here

RESPONSE: Agreed and corrected

Line 73: I wouldn't say this (cite Dave White)

RESPONSE: We reworded this text to "unlike MOST avian brood parasites..." Please note that referee #3 urged us to highlight rather than lessen the importance of the relative ease of the experimental approach with our system, and we have to find a compromise between these two contrasting requests. We now cite White's studies and have also added an appropriate experimental study by Brook and Davies (line 80).

Line 81: very nice experimental set up--you're able to test age vs. experience!!

RESPONSE: We now stress, in addition to experience with the host-parasite interactions, the age difference between our "experienced" and "highly-experienced" treatments, as well as the

additional difference that “highly-experienced” catfish were raised in the host cichlid buccal cavity (naturally) while “experienced” catfish were raised *in vitro* (lines 103 and 323).

Line 114: if juvenile catfish feed on host eggs, then why do the adults do it too? Interesting question to me.

RESPONSE: Our current data cannot provide any insight into this question, but given that groups of catfish participate in spawning, it could represent competition between unrelated individuals (not all adult catfish reproduce during intrusions).

Line 149: post hoc tests between 1 vs. 7 year experimental groups, too? Should be reporting on those, too.

RESPONSE: This is likely a misunderstanding (cuckoo catfish at the age of 1 year are not sexually mature). Our catfish treatments were 5 years (naïve and experienced catfish) and 7 years old (highly-experienced catfish). It was 1 year of coexistence with cichlids for “experienced” (5 year old) catfish. We have reworded the text to avoid this misunderstanding, and we provide details on pairwise comparisons throughout our results (and supplementary tables).

Line 174: actually the host was defenceless in this experiment (since you used *burtoni*, which is not coevolved with the catfish). Please rephrase this sentence.

RESPONSE: We have reworded this statement for greater clarity. The host is actually not “defenceless” because they respond to the threat of egg predation as the cuckoo catfish approach. This is a generalized response, as brood predation during spawning can come from many sources. In contrast, *A. burtoni* hosts do not learn to discriminate catfish eggs once laid (or once collected by the host female and inside her buccal cavity). Actually, re-analysis of our data as prompted by referee 3 showed that hosts appear to slightly improve their responses (visualized in Fig. 2).

We agree that the sentence was partly ambiguous and we now state: “...we demonstrate that brood parasites learn to improve their abilities to successfully parasitize their hosts.” (line 297-8)

Line 199: naked p value. You always need to show the test-statistics, please, not just the p value.

RESPONSE: Corrected, fully referred results of the statistical test are now presented (line 326-7).

Line 297: why not use a Mann-Whitney test here?

RESPONSE: We have re-analyzed our results based on referee 3 advice and use a GLMM to compare the three treatments separately (and hence there is no need to use a Mann-Whitney test to support pooling of experienced and highly-experienced catfish treatments for this analysis).

Line 301: what if you didn't combine it and used n=6 for one of the groups?

RESPONSE: We have done so accordingly and this type of analysis is reported for all behavioural data (i.e. 3 treatments, no pooling).

Line 331: again, need a definition for learning vs. experience.

RESPONSE: We define learning in this version of the ms (line 36-7), as suggested by the referee earlier in their comments (learning, as improvement in response, comes from experience/repeated exposure to the stimulus).

Figs 1-2. there is something very wrong with the color scheme--the naive group should be in blue but it's indicated as red???

RESPONSE: Corrected. We apologize, this was indeed an oversight– it is correct in the captions but incorrect in the in-figure labels.

Fig. 3--please show, analyze, and discuss these same patterns for the two experienced treatment groups, too. Essential!

RESPONSE: We now provide a more detailed account of this pattern for all three treatments, visualized as Fig. 2.

Please note that this point was addressed (although in a less detailed way) in Fig. 1a (model-predicted values for prevalence). So, these patterns were analyzed for all three treatments (results in the first paragraph of results).

Fig. 4: please indicate significant differences with a "*".

RESPONSE: Corrected.

Reviewer #2 (Remarks to the Author):

This study demonstrates that brood parasitic catfish that have more experience in reproducing with cichlid hosts have higher reproductive success than host-naïve catfish. The study is a good read and the methods are sound. The results are clear and interesting, although one has to admit that they are also not particularly surprising given that higher experience with reproduction enhances reproductive success and efficiency in many species. I have a number of specific points and questions below.

Line 44: I do not entirely agree. Imprinting is also a form of learning, and here we know at least that it happens, and that it does in a specialized brood parasite. The learning of habitat features early in life (habitat imprinting) has been shown in an avian brood parasite (Kolecek et al. 2020, Science of Nature; Teuschl et al. 1998, Anim Behav.)

RESPONSE: We now see that our text was not sufficiently specific. We have reworded this paragraph and have added information on habitat imprinting (line 51).

If *A. burtoni* were naïve and had no adaptations to fend off parasites. Are *A. burtoni* ever used as hosts in nature? And, while I can understand the reasons for using naïve hosts, the question is how generalizable the results are for natural systems, in which hosts will have experience with catfish. How relevant is the learning of social cues or timing by catfish for their success in naturally parasitized systems ?

RESPONSE: There are several aspects that we now clarify in the text.

1. Failure to respond to parasitism in *A. burtoni* does not mean that they do not defend themselves from the cuckoo catfish intrusions. They express generalized defence as a part of anti-predatory behaviour (cuckoo catfish are also egg predators). They do not, however, discriminate against catfish eggs (and therefore were chosen to provide an unbiased estimate of parasitism rate, which does not decrease with time in the host buccal cavity). This point is now made more explicitly on lines 94-95. Please note that over time they demonstrate a plastic response towards avoiding catfish more successfully (two “experienced” treatments).

2. We have chosen this host species to examine one specific part of brood parasitism. We agree that the situation in the natural setting is more complex. We agree that conducting experimental work is always a dissection of natural complexity to ask a particular question.

3. *A. burtoni* have not been confirmed as a cuckoo catfish host in the wild – but there were only two studies completed (Sato 1986 and our unpublished data), along with one additional published incidental report (Takahshi and Koblmüller 2020). *A. burtoni* has a parapatric distribution with the cuckoo catfish, being typical inhabitants of river mouths, where they may overlap with the cuckoo catfish. Indirect evidence, however, also suggests that *A. burtoni* are not hosts of the cuckoo catfish (i.e. failure to respond to previous parasitism compared to its ability in the confirmed host species).

As the methods come afterwards, it would help to have some basic information here:

-line 82: How many catfish in each treatment?

RESPONSE: Added as requested (lines 100-105), always n = 36 catfish per treatment.

-Line 86/87: How many male and female catfish and cichlids per tank, and how many tanks per treatment? Up to this point one may think there was only one tank per treatment, but later it becomes clear there were 6 of them

RESPONSE: Added as requested (lines 107-109).

line 88: 'strong reciprocity' is a term used in the cooperation/altruism literature, so please rephrase to avoid misunderstandings

RESPONSE: Thank you for pointing out this issue. We reworded it as 'interactions' (line 83).

line 117: was predation confirmed by video analysis? Could there be alternative explanations of reasons why the cichlids had different clutch sizes?

RESPONSE: Yes, predation is very obvious throughout the recorded spawnings, and in some cases predation was substantial. We now include a supplementary video (and refer to it in the text) to demonstrate predation.

How was the number of catfish eggs corrected for number of host spawning events – see also my question below about the number of cichlid spawnings in the treatments. If that differed systematically between treatments, part of the results may not be explained by catfish learning but by the chances of catfish to disseminate their available eggs. If, as an extreme example, in one treatment twice as many spawnings occurred than in another, but catfish have only a certain amount of mature eggs to spend, then the proportion of parasitized clutches, the number of eggs across cichlid clutches (and possibly even the catfish clutch size) would be affected.

RESPONSE: The number of spawnings was not identical (n = 146, 174, 191 per treatment), but was high overall and did not differ significantly or systematically among treatments. This is now specifically articulated in the paper (lines 180-181) and statistically analyzed in the methods (lines 378-384). The highest number of spawnings was in the treatment with naïve catfish (where also the host egg number was highest), suggesting that the outlined scenario (more eggs per clutch when possibilities to parasitize were lower) did not play a role in our experiment.

In addition to Fig. 1 the raw data points should be plotted to see how the interaction comes about, and whether the data are equally distributed along the x-axes, or whether there are time patterns.

RESPONSE: We now fully revised Fig. 1 to address this point, which has enabled us to plot the raw data in a visually informative presentation. We have also included observed data points in the original version of Fig. 1 (with temporal trends) which has been relegated to the supplementary material (as Suppl. Fig. 1) to avoid repetition of data presented in Fig. 1 and 2, since another request was for presenting temporal trends with the cumulative number of spawnings for all three treatments. The new Fig. 2 provides information that is requested here and visualizes that there

are no time patterns in the data, with spawnings distributed equitably across time in all three treatments.

Even if not significant, the success of the experienced catfish declines over the course of the experiment, which in part drives the significant interaction. How does this come about, if *A. burtoni* hosts really do not learn?

RESPONSE: Thank you for bringing up this point. Indeed, there is a decrease in catfish success for experienced and highly experienced groups over the course of our experiments. We have reworded the text accordingly (lines 164-166).

To address this important issue, we have additionally tested whether temporal increase in parasite success of naïve catfish holds irrespective of whether it was compared against other treatments or in a treatment-specific analyses against the null expectation of no temporal trend. This rapid increase (slope of the regression) was highly significant irrespective of the other two treatments. We present this on lines 167-169 and the numerical results are detailed in Supplementary Tables 5 and 6.

Please correct the in-figure labels of Fig. 2, which say that blue is highly experienced. I assume the legend is correct though, which claims that blue is naïve, and so it fits the results.

RESPONSE: Corrected. Thank you for pointing out to this oversight. Please note that figures are reworked overall to meet various reviewers' requests.

Fig 4: should be separated by experienced and highly experienced to avoid the argument that the differences are merely because of the higher age of the highly experienced catfish.

RESPONSE: We were initially reluctant to separate "experienced" and "highly-experienced" treatments given the low number of successfully videorecorded replicates of the latter. However, following requests from referees, we have now split them for analytic and visual presentation.

While we interpret them with care (given sample size limitations), they demonstrate the same outcome, except that the number of catfish individuals intruding host spawning was not higher in highly-experienced catfish when compared to naïve catfish (in contrast to experienced and naïve catfish treatments). This is presented on lines 220-222, and illustrated visually in Fig. 4c.

line 134: sample size of subset?

RESPONSE: We now state the sample sizes for the subset of analyzed cichlid spawnings (lines 190-191).

line 146: explain "T-position" for non-cichlid readers

RESPONSE: We clarified this term, including detailed description of what is represented by 'male' and 'female' T-position (line 67, 199, 214-5, 359-361). We also provide a supplementary video where each described behaviour is presented (with subtitles pointing to each particular behavioural unit).

line 167: check reference numbers. ref 26, for instance is "26. Materials and methods are available in the supplementary materials." which does not seem to fit here

RESPONSE: We have double-checked all references. We apologize for this mistake which arose during copy editing of a previous version.

line 172: is there any evidence for evolutionary traps in this system?

RESPONSE: No, there is no evidence. Unfortunately, our knowledge on fish brood parasitism is still too fragmentary to confirm whether cuckoo catfish specialize on particular host species.

We cite “evolutionary trap” in the final paragraph, referring to brood parasitism in general. Please note that we have rewritten the discussion substantially.

In the methods, please give the exact number of host spawnings in each treatment, and please test if there was an effect of treatment on number of cichlid spawning

RESPONSE: We added this information (total number of spawnings per treatment, naïve = 191, experienced = 174 spawnings, highly experienced = 146) to the results (lines 180-182), along with formal analysis in methods (lines 379-380) which demonstrates no difference between naïve and experienced treatments ($p = 0.519$) and a non-significant trend between naïve and highly-experienced ($p = 0.070$)

line 269: so what happens when cichlid egg number if taken in the models instead of the proxy of cichlid female size?

RESPONSE: We have used female body size as a covariate because the number of host eggs was affected by catfish predation and was, therefore, the least precise of these two otherwise collinear variables.

line 274: Cichlid spawning activity includes number of spawnings, not only number of eggs. See above – are there treatment differences?

RESPONSE: We have now formally tested this possibility and results are reported on lines 379-360. We saw no difference between naïve and experienced catfish ($p = 0.519$) and a non-significant trend in the comparison between naïve and highly-exp catfish ($p = 0.070$).

Reviewer #3 (Remarks to the Author):

Review of: Learning in brood parasites: experience as a key to success for the cuckoo catfish

While there is a fair bit of work in the avian brood parasitism literature on the role that information and learning plays in refining host defences, understanding the role that learning plays in refining the behavior of brood parasites is something that has basically gone unstudied, because the systems are just not well suited for this kind of work. The authors here have used the practical and logistical advantages of their unique fish system to experimentally investigate how learning refines parasitic behaviours, which is great and in my opinion worthy of serious consideration for publication in Nature Communications.

That being said, the execution of the manuscript is (frankly) pretty ordinary, which made this a very interesting paper to review. Overall, given that the actual study is very novel and the execution of the experiments is suitable, the paper should be given a chance for resubmission, as these results should be published in a top-tier journal. Of course, that can't happen if the manuscript isn't good enough, so a very serious overhaul is needed.

In particular, my main issue(s) all seem to stem from the same key issue: the frivolous use of language and the generally uninformed pitch of the paper seems to stem from a (seemingly very) superficial understanding of the non-catfish related brood parasitism (and broader coevolution) literature. I think the most upfront evidence of this is the stark contrast between the thoroughness of the referencing and detailed writing with which the authors discussed catfish literature (e.g.

paragraph starting L 55) compared to the previous paragraphs of the introduction, and there was an equally superficial discussion of the literature and how the results fell within it and added to it in the results/discussion section. I have given some examples below, but generally a much more thorough knowledge of the relevant literature (i.e. you're pitching this paper as a general coevolution/brood parasitism paper – you need to know this literature) and a more thoughtful placement of this result within this context (including discussion) is needed.

I sincerely think that this result is suitable for publication in Nature Communications, and hope that the manuscript is given an opportunity for resubmission.

RESPONSE: We thank the referee for positive views on the importance of our study, and for constructive criticism throughout. We have tried to add more specific notes and explanations for text referring to general coevolution and brood parasitism. We acknowledge that this text was originally prepared for a journal with a more stringent word count limit. We have rewritten abstract and introduction, added more analyses of our data, reworked the figures, and added more discussion.

General writing:

Abstract

L 14-15: It's true that evidence (especially direct evidence) of learning in brood parasites is very rare, but saying the role of "other factors... is poorly understood" is untrue. There is a lot of literature on the role that social and personal information have on host behaviours, at least in birds, as well as the effects other factors have in shaping these interactions, so I would tone down this kind of language.

RESPONSE: We have changed from "other factors...poorly understood" to "role of learning...can be masked", with a similar statements further in the main text.

L18-19: This is not true. Presumably you mean interspecific brood parasitism (brood parasitism is too vague on its own for a statement like this – probably also applies to L13), but even so, interspecific brood parasitism has recently been described in coral reef fishes (e.g. Tarel et al. 2020 Molecular Ecology <https://doi.org/10.1111/mec.15243>)

RESPONSE: We apologize for the lack of precision in our expression. We have meant OBLIGATE brood parasites. We added the term "obligate" which automatically excludes intra-specific brood parasitism as well as facultative parasitism reported in some other fish species.

L 23: Please change "superior" to "improved" (or something like this).

RESPONSE: Corrected as suggested.

L 23-25: The language here does not sit well with me – this is not the first study to demonstrate that an arms race occurs throughout the lifespan of brood parasites and hosts, but you probably (to my knowledge) could argue that you are providing the first direct evidence (or at least among the first) of learning in brood parasites with regards to their interactions with hosts.

RESPONSE: We have reworded the sentence, also in response to other referees' suggestions (who did not like reference to "arms-races" for individual lifetime), along with constraint of the word count limit to 150 words). We do not claim to present the first (direct) evidence for within individual "arms race" or "learning of brood parasites to interact with their hosts".

Please note that we specifically refer to additional previous studies in the Introduction.

Introduction

L 26-27: I don't really see the point of this first sentence – parasitism is successful because parasites are abundant? In my opinion that's not really an opening sentence for a journal like Nature

Communications. Generally, this problem (lots of words not saying much) is pretty ubiquitous in the introduction – more thorough incorporation of the literature is needed.

RESPONSE: We understand that opening sentence is a matter of taste. We have completely reworded the opening sentence.

L 27-28: "Particularly costly" compared to what, and if so, why isn't it referenced? This seems a very frivolous or careless use of language and is something needs to be addressed.

RESPONSE: We reworded "costly" for "valuable" and provide a reference for this statement (that parental care is valuable/costly). The 'cost' of parental care can be compared from multiple perspectives (energetics, time, increased predation risk, etc.), but we believe that further elaboration is outside the scope of this general statement.

L 29: "rapid evolution of host counter-adaptations" should be referenced (maybe Spottiswoode & Stevens 2012 Am Nat would be an appropriate reference here).

RESPONSE: Thank you for your suggestion. We now cite this paper and also use a recent review by Medina et al., 2020 to underpin this statement (see lines 28-29).

L 34: Is the word "rapid" really needed? I understand it makes it sound more urgent, maybe, but I think "coevolutionary dynamics" is fine, and more accurate. This should also be referenced (maybe a review or two – there are plenty of great ones to choose from).

RESPONSE: "We agree that "rapid" is not needed and was deleted. Two references have also been added to support the statement.

L 36-37: This is not true, and disregards a very substantial literature on the topic, at least in birds (see work by Campobello, Davies, Feeney, Langmore, and references within their papers).

RESPONSE: We see that our statement was not succinct. It was not our intention to state that there is no existing literature on learning in the host – brood parasite relationship. We have elaborated on this point in the following sentences where we refer to the studies of Feeney & Langmore and others. We have reworded this sentence to make clear that research in this field is mostly host-biased and the parasite's perspective is relatively understudied. Further we also acknowledge the studies of Campobello & Sealy and Davies & Welbergen (and others) in the following text (10 references to topical studies cited in this paragraph).

L 41: I do not think that the terms "parasitic skills" and "master their abilities" are appropriate language for a journal like this.

RESPONSE: We have replaced "sophisticated parasitic skills" for "complex behavioral repertoire" and we avoid the use of the word "skill" throughout the text (replaced for "ability" or "capability").

L 43-45: Where are the references to support this statement? There is also a lot of information on the life history traits that brood parasites (not just cowbirds) target, which is relevant here.

RESPONSE: We have added supporting papers to the first sentence of this part (note that we have referred to individual studies in the following sentence). We have also replaced "all" for "most" (line 45), and refer to the European cuckoo which is capable of imprinting on natal habitat, in addition to cowbird studies (lines 51-53).

L 53: This is not always true, see work by Gloag and/or Soler. Also probably looking up mafia hypothesis work.

RESPONSE: Yes, we agree that our statements were too conclusive. We now state: "...OFTEN use stealth..." and "...MAY BE difficult to follow." (line 58)

Paragraph starting L 55: This is the first paragraph that I have felt is of the quality expected for a journal like Nature Communications. The author should note how superficial and sparsely referenced the previous paragraphs are compared to this. It appears as though, while they certainly know their stuff regarding this system, they are less aware of the other relevant literature on the topic, which is very clear in their writing. I suggest a major overhaul of the majority of the introduction, and a more thoughtful use of language and care used when developing the context for this work.

RESPONSE: Thank you for this constructive criticism. We have comprehensively reworded the introduction and added more specific references. We hope that the current version is clearer and more specific. We acknowledge that the previous version was prepared for a journal with much more stringent word (and reference) count limits. Please note that, overall, we had to accommodate comments from three referees that were sometimes partly conflicting.

Results & discussion

It is very difficult to review the stats when they are (sometimes) in the figure captions. I am not sure if Nature Comms accepts this format, but for readability, I find it much more helpful having the stats in the manuscript text.

RESPONSE: We now adapt the presentation of statistics to Nature Communication format and all results are referenced in the main text, with full results in Supplementary Tables and complete annotated R code accessible on Figshare along with the primary data.

L 101: Please change "superior parasitic skills" to something more appropriate.

RESPONSE: The term "skills" is not used in the current version of the ms at all. We use "lower/greater parasitism success", and later in the sentence replaced "skills" with "must learn to gain the experience".

L 102: Please try and get rid of the word "skills" its vague and not very helpful. It also just sounds a bit silly.

RESPONSE: The word "skill" is not used in the revised version of the ms.

L 103-104: Percentages are not particular informative here by themselves and I'd suggest providing N wherever possible as well. Also, you should be providing more information with your stats (e.g. chi2 and df with your glmm, or whatever depending on the model). I would also suggest providing an R Markdown so that we can check the stats more thoroughly.

RESPONSE: Thank you for this suggestion. We use percentages here, because the number of spawnings in each treatment group was comparable, but not equal (n = 191, 174, 146). Percentages therefore allow direct comparison between treatments, while counts would not.

We now provide more detail on the statistics in the main text (with full details in Supplementary Tables) and full (annotated) R script for our analyses (previous version included only final models).

L114: Parasite experience, or experience with a parasite?

RESPONSE: We agree that this is important point. We think it is indeed "parasite experience", because clutch size is reduced by catfish predation, not by host decisions.

To avoid any ambiguity, we reworded this opening sentence for the section to read: "Experienced (and highly-experienced) cuckoo catfish decreased the size of host clutches." (line 174)

L 130-133: It's worth noting that other studies in birds (albeit on hosts) have found that learning can also be extremely quick (Langmore et al. 2012 Behav Ecol, Feeney & Langmore 2013 Bio Lett), and this study is interesting because you're finding similar evidence in the parasite (which is something we just can't really do in birds).

RESPONSE: We have added the following sentence: "Similarly rapid learning was previously detected in avian hosts (Feeney & Langmore 2013)" (line 267). Note that we did not find a temporal component of learning in Langmore et al. 2012 and therefore we cannot cite it here.

L139: The results are just very inconsistently presented. Here, the stats are presented differently and more correctly than they are above (L104) – why? Also, why are some stats presented in figure caption, but others are in text? It just seems like the paper is a bit of an unpolished draft, rather than a polished manuscript ready for submission to a major journal, which is a bit irritating, because your result is so interesting.

RESPONSE: We now consistently refer to all statistics in the main text. In the previous version, figure captions were used for some statistical results (i.e. those which were visualized), but obviously not for analyses which were not presented visually as figures (and then we used full reference to the stats in the text).

L 143-151: You should probably include references to supplementary videos and/or screenshots throughout this paragraph.

RESPONSE: Thank you for this suggestion. We have uploaded a video in which we describe (with captions) all relevant behaviours referred to in the main text.

Paragraph starting L 161: This is way too superficial (as is the rest of the discussion parts of this results/discussion section). Much more insightful and thorough consideration of the brood parasitism literature is required, how this study fits in, and the broader implications of this result. This requires substantial work, as it is currently no where near the quality I would expect for a journal like this. That being said I think you can fix it, but I would almost suggest reaching out to someone who works on this to help, as it currently falls (very) short.

RESPONSE: We have now expanded the discussion, reviewed by M. Honza who works on avian brood parasitism (see Acknowledgements).

L 168: I mean, your fish system is just more appropriate for examining this, because you can put them in tanks. You should just say this, because its true.

RESPONSE: We prefer not to state this point explicitly, given that referee 1 in particular, referred to experimental work by D. White and M. Taborsky. The appeal of referee 1 directly interferes with this request to specifically state that our system is more amenable for experimental work in captivity than avian brood parasitism.

Methods

Generally seem fine and well written. I would strongly suggest submitting an R Markdown file with all analyses so that we can more easily assess the suitability and execution of the stats.

RESPONSE: We include annotated R script as a part of our primary data file upload (Figshare Repository).

We would like to thank all three referees for constructive criticism, which we believe has clarified errors and ambiguities in our work. We would also like to thank the editor for inviting us to revise our submission.

Reviewers' Comments:

Reviewer #3:

Remarks to the Author:

2nd review of "Learning in brood parasites: experience as a key to success for the cuckoo catfish"

I think the authors have done a really good job editing this paper. I have a few minor points, but I am happy with how they addressed my comments, and it seems like they've done a good job of addressing those by the other reviewers. I am really glad, because I think this is a unique result and I think it should be published in a reputable journal.

Well done! Pleasure to review, and looking forward to seeing it published.

L14: I am happy to be convinced otherwise, but my thoughts are that in this system, the trajectory of a coevolutionary interaction is affected by learning, rather than learning happening in addition to a coevolutionary interaction. Personally, I find this way of thinking about it (coevolution and learning as distinct things) a bit strange, and would suggest modifying the text accordingly. I would resist suggesting that learning happens in addition to coevolution, personally.

L15-16: I don't know if the word "masked" is correct here. It's definitely more difficult to detect, but that's different to it being masked. I would suggest changing the phrasing here.

L 22-23: same as L14. Learning is part of the interaction, in my opinion.

L28 CAN elicit

L 83: amenable to LAB-BASED experimental studies

L 115: If this section of the manuscript is the introduction (presumably, but there is no heading), I do not understand why this paragraph is here. It is a summary of the results, immediately before the results section. It reads more like the first paragraph of a discussion (just missing the opening and closing sentences). I would suggest deleting this paragraph entirely.

REVIEWERS' COMMENTS

Reviewer #3 (Remarks to the Author):

2nd review of “Learning in brood parasites: experience as a key to success for the cuckoo catfish”

I think the authors have done a really good job editing this paper. I have a few minor points, but I am happy with how they addressed my comments, and it seems like they’ve done a good job of addressing those by the other reviewers. I am really glad, because I think this is a unique result and I think it should be published in a reputable journal.

Well done! Pleasure to review, and looking forward to seeing it published.

RESPONSE: We thank for positive evaluation and insightful critical comments.

L14: I am happy to be convinced otherwise, but my thoughts are that in this system, the trajectory of a coevolutionary interaction is affected by learning, rather than learning happening in addition to a coevolutionary interaction. Personally, I find this way of thinking about it (coevolution and learning as distinct things) a bit strange, and would suggest modifying the text accordingly. I would resist suggesting that learning happens in addition to coevolution, personally.

RESPONSE: Edited as suggested (“In addition to coevolution” deleted and “also” added).

L15-16: I don’t know if the word “masked” is correct here. It’s definitely more difficult to detect, but that’s different to it being masked. I would suggest changing the phrasing here.

RESPONSE: Changed to “concealed”.

L 22-23: same as L14. Learning is part of the interaction, in my opinion.

RESPONSE: Edited as suggested. “in addition to” changed to “within the” ...coevolutionary arms races...).

L28 CAN elicit

RESPONSE: Corrected.

L 83: amenable to LAB-BASED experimental studies

RESPONSE: Corrected to “laboratory-based”.

L 115: If this section of the manuscript is the introduction (presumably, but there is no heading), I do not understand why this paragraph is here. It is a summary of the results, immediately before the results section. It reads more like the first paragraph of a discussion

(just missing the opening and closing sentences). I would suggest deleting this paragraph entirely.

RESPONSE: This section was retained as this follows general editorial guidelines, as well as direct appeal of the editor.

We would like to thank the referee again for very thorough review of our paper.